# Design and Energy Requirements of a Photovoltaic-Thermal Powered Water Desalination Plant for the Middle East

**DOI:** 10.3390/ijerph18031001

**Published:** 2021-01-23

**Authors:** Saeed Alqaed, Jawed Mustafa, Fahad Awjah Almehmadi

**Affiliations:** 1Department of Mechanical Engineering, College of Engineering, Najran University, King Abdulaziz Road 1988, Najran 61441, Saudi Arabia; saalqaed@nu.edu.sa; 2College of Engineering, Muzahimiyah Branch, King Saud University, P.O. Box 800, Riyadh 11421, Saudi Arabia; falmehmadi@ksu.edu.sa

**Keywords:** PV-thermal, desalination, hybrid, solar collector, optimization, middle east

## Abstract

Seawater or brackish water desalination is largely powered by fossil fuels, raising concerns about greenhouse gas emissions, particularly in the arid Middle East region. Many steps have been taken to implement solar resources to this issue; however, all attempts for all processing were concentrated on solar to electric conversion. To address these challenges, a small-scale reverse-osmosis (RO) desalination system that is in part powered by hybrid photovoltaic/thermal (PVT) solar collectors appropriate for a remote community in the Kingdom of Saudi Arabia (KSA) was designed and its power requirements calculated. This system provides both electricity to the pumps and low-temperature thermal energy to pre-heat the feedwater to reduce its viscosity, and thus to reduce the required pumping energy for the RO process and for transporting the feedwater. Results show that both thermal and electrical energy storage, along with conventional backup power, is necessary to operate the RO continuously and utilize all of the renewable energy collected by the PVT. A cost-optimal sizing of the PVT system is developed. It displays for a specific case that the hybrid PVT RO system employs 70% renewable energy while delivering desalinized water for a cost that is 18% less than the annual cost for driving the plant with 100% conventional electricity and no pre-heating of the feedwater. The design allows for the sizing of the components to achieve minimum cost at any desired level of renewable energy penetration.

## 1. Introduction

As global demand for freshwater increases, desalination technology becomes indispensable because natural supplies of fresh water are limited. Currently, the world population affected by water shortages is nearly 40% [1]. The current global desalination capacity is 90 million m^3^ per day, far too small to offset these shortages [2]. Most of the desalination power plants are concentrated in the Middle East, where dry Arab countries use desalination to meet their fresh water demands [3].

The Kingdom of Saudi Arabia (KSA) is the largest producer of desalinized seawater, with 17% (17.2 million m^3^ per day) of the worldwide capacity. Over the next 20 years it is predicted that the KSA will need to increase this output by 6 million m^3^ per day [4]. Burning oil supplies creates the energy needed for this desalination process. The KSA currently consumes more than 1.5 million barrels of oil per day for this purpose [5]. A barrel of oil has enough energy to produce about 5 m^3^ of desalinized water [4]. The fossil-fuel energy consumed for desalination raises concerns about greenhouse gas (GHG) emissions. Before 2016, electricity from oil production was very low, effectively subsidized by the government, which started to reduce the support gradually. The use of renewable energy for powering desalination is therefore of great interest, although currently, only about 0.02% of the world’s freshwater production from desalination is renewable-powered [6]. Solar photovoltaic (PV) or solar thermal represents most of this fraction, and these technologies are appropriate for the Middle East due to the abundant solar resources. Consideration by the Persian Gulf countries to move to sustainable renewably powered solar desalination is a current effort. However, the technology is presently expensive relative to burning oil, and it can be difficult to implement. 

In the KSA, desalination plants are dual-purpose, producing both electricity and water for coastal urban centers. To avoid transportation costs, large-scale plants are mostly located on the Red Sea or the Persian Gulf, where there is both an ample supply of seawater and dense populations. Inland KSA cities sometimes incur a high cost for pumping water from coastal plants. To avoid these pumping costs, isolated cities or facilities such as hotels, hospitals, offshore platforms, ships, etc., can use small-scale desalination plants (mobile or stationary) to supply fresh water. In remote communities that have periodic water shortages, small mobile desalination units are often used. There are two basic types of desalination: thermal and membrane. Globally, 68% of desalination is membrane-based, 30% is thermal, and the remainder uses other processes [7]. Within the thermal desalination sector, multi-stage flash (MSF) technology is the most common, and reverse-osmosis (RO) is the most common membrane process [8]. MSF operates by passing heated seawater into a sequence of vacuum chambers, to promote flash evaporation. The resulting vapour condenses onto heat exchangers, which transfer thermal energy to the incoming feed seawater. Because MSF is reliable and easy to operate and can be powered by burning fossil fuels, it is frequently used in Middle Eastern nations [3]. The performance of an MSF system depends on many factors, such as the temperature of the feedwater [8]. If the feedwater temperature is too low, then the salinity of the product water is too high [9]. This creates a seasonal effect on the quality of water produced from this process. Researchers have concluded that production is higher in summer than in winter due to the higher summer feedwater temperatures [10]. 

RO operates by allowing the passage of water molecules, excluding salts and other impurities, through special membranes by applying high pressure. RO is the most common membrane process because of its low energy requirements and greater recovery percentage [11]. RO desalination is also affected by feedwater temperature, including the recovery ratio, consumed energy per cubic meter, and salt rejection [12]. Higher feedwater temperatures have been shown to improve the recovery ratio and reduce power consumption [13]. MSF requires a mix of thermal and electric energy (for pumping), whereas RO primarily requires electrical energy. However, if the RO feedwater requires heating to improve performance, then RO also requires thermal energy. Solar energy conversion into thermal or electric energy is credible and is the most prevalent renewable energy source in the Middle East. Consequently, the greatest attention for renewable desalination has focused on solar-based systems. Renewable desalination is appropriate for dry, sunny, and remote regions where no other mode of power is possible. Photovoltaic (PV) cells convert solar radiation into electrical power, but their efficiency reduces when their temperature increases. One way to overcome this problem is to cool the solar cells with a circulating flow of coolant, which improves electrical efficiency while producing thermal energy. The result is a hybrid photovoltaic/thermal (PVT) unit [14,15,16,17]. Because electricity production is most often the priority, PVT is not widely used as a PV or a solar collector units alone. However, PVT systems have a lower cost per unit of electricity and heat produced for the same total surface area needed for their installation [15]. The total requirement area for a PVT collector system is about 40% less than separate PV and solar thermal collectors with the same total capacity. In many applications of hybrid PVT systems, the prioritization of electrical output is as such that the operating conditions of the heat transfer unit are controlled, thereby maximizing electrical output, not thermal output. However, PVT designed to optimize heat transfer is possible, creating higher outlet fluid temperatures while sacrificing some PV efficiency [18]. Therefore, PVT is very attractive to numerous applications because it is flexible in terms of thermal versus electric energy outputs. Besides, PVT could be appropriate for desalination applications in KSA, where high ambient and operating temperatures reduce the efficiency of PV systems. 

The goal of the research described herein is to optimize the integration of PVT for a small-scale desalination plant appropriate for a remote community in the KSA. The design uses a mix of PVT and conventional electrical power to drive RO desalination. The PVT thermal output can raise the feedwater temperature, thus reducing the viscosity and the power required for pumping, improving RO performance, and the PVT electrical output provides some of the RO power needs. Excess PVT electrical power may be stored in a battery for nighttime desalination, along with grid power as needed, to maintain continuous RO production. The following describes the structure of this paper. First, a detailed illustration of the methodology and the model used in this study is present in Section 2. Second, Section 3 shows the dispatching algorithm for the PVT while Section 4 presents a detailed economic model. An optimization algorithm for the proposed system is in Section 5. The results of the cost objective optimized function are present in Section 6. Lastly, Section 7 summarizes the research outcomes and major findings.

## 2. Methodology

### 2.1. Plant Design 

Figure 1 contains a block diagram of the proposed water desalination plant that combines PVT and RO technology. The feedwater source is assumed to be a reservoir of brackish water. This source is sufficiently large to render a constant mass flow to the system, part of which flows through the PVT array to gain thermal energy and reduce the PV cell temperature in the array. The PVT array tilts to track the sun. At night or during times when the air temperature is too low, the feedwater bypasses around the PVT array so that it does not lose heat to the atmosphere. A fixed volume thermal storage tank is utilized to optimize the benefit of the acquired thermal energy over time. If the storage tank temperature drops below some minimum threshold temperature, auxiliary heating is obtainable. The heated water stored in the tank provides a fixed flow to the RO, and its higher temperature serves to reduce the electrical power needs for various pumps throughout the system because of its lower viscosity. On the electrical side of the system, the PVT array provides as much electrical power as possible for pumping requirements. The excess electrical energy generated during the day is stored in a battery and then used by the system at night. Grid power makes up the remaining electrical needs, especially in the early morning hours, after the battery has drained.

### 2.2. PVT Modeling

Although there are several PVT configurations, the type considered here is a flat-plate collector. The basic structure of this collector displays as a cross-section in Figure 2 where G refer for solar irradiation falling on PVT flat-plat collector. 

Fluid circulates beneath a layer of standard PV cells, extracting heat from the cells and allowing them to operate at a lower temperature with higher electrical efficiency. There are ways to make such a structure favor the electrical or thermal output. For example, using an extra layer of glass on top of the PV cells increases the temperature and improves thermal conversion of the incident solar energy, but somewhat degrades electrical performance [19]. Another option is to use a glazed glass cover to enhance the thermal energy production of the PVT [20]. By arranging the multitude of PVT panels in an optimal series/parallel configuration, the total temperature rises from the entirety of the PVT panels are controllable. The full PVT array used in the design consists of $N_{p}$ parallel-connected strings of $N_{s}$ series-connected collectors [21]. 

### 2.3. Thermal Model

The PVT model equations used to determine thermal and electric energy production are based on conservation of energy principles [20]. The surmised array consists of $N_{p}$ collectors in parallel and $N_{s}$ in series, for a total of $N_{s}\times N_{p}$ panels. The total thermal power transferred by the entire array to the fluid is ${\dot{Q}}_{u,total}$, given as the sum of the heat produced from each of the individual serial paths of panels, as shown.
(1)$${\dot{Q}}_{u,total}=N_{P}*{\dot{Q}}_{u,N_{s}}$$

The quantity ${\dot{Q}}_{u,N_{s}}$ represents the thermal power transferred to the fluid flowing through $N_{s}$ panels in series.
(2)$${\dot{Q}}_{u,N_{s}}=N_{S}A_{C}F_{R}\left[S\left[\frac{1-{\left(1-K_{K}\right)}^{N_{S}}}{N_{S}K_{K}}\right]-U_{L}\left[\frac{1-{\left(1-K_{K}\right)}^{N_{S}}}{N_{S}K_{K}}\right]\left(T_{fi}-T_{a}\right)\right]$$

In this Equation, $A_{C}$ is collector area for each panel, $F_{R}$ is a heat removal factor, S is the incident solar radiation per unit area absorbed by the collector, $U_{L}$ represents the overall thermal conductance for heat losses, $T_{fi}$ is inlet flow temperature, and $T_{a}$ is ambient air temperature. The quantity $K_{K}$, is physically given by
(3)$$K_{K}=\frac{A_{C}F_{R}U_{L}}{mC_{p}}$$
where m is the fluid mass flow rate and $C_{p}$ is the fluid specific heat. The solar radiation per unit area absorbed by an individual PVT panel is
(4)$$S=I\left(\alpha \tau \right)\left(1-\eta_{c}/\alpha \right)$$
where I is solar irradiation intensity, $\eta_{c}$ is electrical efficiency at cell temperature, α is the absorptance of the collector, and τ is the transmittance of the glazing on the collector. The outlet fluid temperature from $N_{s}$ panels in series are given by
(5)$$T_{fo}=\left[\frac{S}{U_{L}}+T_{a}\right]\left[1-e^{\left(\frac{-N_{s}A_{C}F_{R}U_{L}}{mC_{p}}\right)}\right]+T_{fi}e^{\left(\frac{-N_{s}A_{C}F_{R}U_{L}}{mC_{p}}\right)}$$

The collector heat removal factor is given by the following Equation,
(6)$$F_{R}=\frac{mc_{P}}{U_{L}}\left[1-e^{-U_{L}F^{\prime }/mC_{p}}\right]$$
where $F^{\prime }$ is a constant collector efficiency factor. The PV/T cell temperature $T_{C}$ is calculated using the following Equation: (7)$$T_{C}=\frac{I\left(\alpha \tau \right)+U_{tc,a}T_{a}+h_{c,p}T_{P}}{U_{tc,a}+h_{c,p}}$$
where $T_{P}$ is plate temperature, and $U_{tc,a}$ and $h_{c,p}$ are taken from Duffie and Beckman [22], Tiwari [23], and Tiwari and Sodha [24]. Finally, the PVT array thermal efficiency defines as the ratio of thermal energy transferred to the fluid to the total input of solar energy on the array.
(8)$$\eta_{th}=\frac{{\dot{Q}}_{u,total}}{N_{S}N_{P}A_{C}I}$$

### 2.4. Electrical Equations

The electrical power from the PVT panel is given by
(9)$$Q_{e,total}=Q_{e}N_{s}N_{P}$$
(10)$$Q_{e}=\frac{A_{c}S\eta_{a}}{\alpha }\left\{1-\frac{\eta_{r}\beta_{r}}{\eta_{c}}\left[F_{R}\left(T_{fi}-T_{a}\right)+\frac{S}{U_{L}}\left(1-F_{R}\right)\right]\right\}$$
Here the second term accounts for the reduction in electrical energy conversion due to an elevated cell temperature (calculable from Equation (7)). 

The electrical efficiency of the PV/T panel is given by:(11)$$\eta_{c}=\eta_{r}\left[1-\beta_{r}\left(T_{C}-25\right)\right]$$
where $\eta_{r}$ is the electrical efficiency for a cell temperature of 25 C, and $\beta_{r}$ is a temperature coefficient for solar cell efficiency, with a value of $\beta_{r}=0.0045\;1/C^{\circ }$.

### 2.5. Solar Irradiation, Ambient Temperature, and Feed-Water Temperature

Typical meteorological year (TMY) data for a specific location are used to provide hourly solar irradiation I and ambient temperature $T_{a}$. The solar irradiation is generated assuming that each panel in the array tilts on a fixed axis to track the angle of the sun [25]. The effect of wind on the heat transfer from the PVT array is ignored. Before being pumped through the PVT array, the feedwater is pumped out of the ground and into a holding reservoir. In this case, the feed-water temperature $T_{fi}$ is approximated by low-pass filtering the ambient air temperatures, such that the feed-water temperature tracks the low-frequency variations in air temperature. Figure 3 shows a plot of air temperature from a TMY data file and the filtered air temperature as a model for feed-water temperature. This allows for modelling the effects of the large thermal mass of the feed-water reservoir. Using the solar irradiation, ambient air temperature, and feed-water temperature, the useful thermal and electrical PVT outputs are computed from (1) to (11).

### 2.6. Tank Temperature Modeling

A direct (open) configuration for the storage tank system is defined such that the water in the tank does not circulate through the PVT array. Instead, some of the brackish source water passes through the PVT array to gain heat and then flows directly to the storage tank. At night, or when the air temperature is too low, the source water bypasses the PVT array and flows directly to the tank. The total volume of feedwater entering and leaving the tank at each time interval is constant. The temperature in the tank therefore fluctuates according to the available solar energy. Auxiliary heating is used to maintain the tank temperature above a specified temperature $T_{min}$. 

The following energy balance equation utilized governs the tank and determines the tank temperature variations in time.
(12)$$M_{T}C_{p}\frac{dT_{T}}{dt}={\dot{M}}_{f}C_{p}\left(T_{fi}-T_{T}\right)+{\dot{Q}}_{u,total}+{\dot{Q}}_{aux}+{\dot{Q}}_{loss}$$

The tank mass $M_{T}$, feedwater mass flow ${\dot{M}}_{f,}$ and heat capacity $C_{p}$ are constants in this Equation. 

The useful heat from the PVT array that is passed into the tank ${\dot{Q}}_{u,total}$ is determined using Equation (1). Auxiliary heat ${\dot{Q}}_{aux}$ is added to the tank to maintain its temperature above $T_{min}$, and the thermal losses from the tank are given by ${\dot{Q}}_{loss}$. These quantities are given as follows:(13)$${\dot{Q}}_{aux}=\{\begin{array}{c} 0\;\;\;\;\;\;\;\;if\;T_{T}\geq T_{min}\\ M_{T}C_{p}\left(T_{min}-T_{T}\right)\;\;\;if\;T_{T}<T_{min} \end{array}$$
(14)$${\dot{Q}}_{loss}=U_{T}A_{T}\left[T_{a}-T_{T}\right]$$

The tank surface area is $A_{T}$ and $U_{T}$ is a constant loss coefficient. Equation (12) is discretized by approximating the derivative as:(15)$$\frac{dT_{T}}{dt}\approx \frac{T_{T}\left(t+\Delta t\right)-T_{T}\left(t\right)}{\Delta T}$$

Making this substitution into Equation (12) and solving for $T_{T}\left(t+\Delta t\right)$ gives the following update equation for the tank temperature:(16)$$T_{T}\left(t+\Delta t\right)=T_{T}\left(t\right)+\frac{{\dot{Q}}_{u,\mathrm{total}}\left(t\right)+{\dot{Q}}_{aux}\left(t\right)+{\dot{Q}}_{loss}\left(t\right)+{\dot{M}}_{f}C_{p}\left[T_{fi}\left(t\right)-T_{T}\left(t\right)\right]}{M_{T}C_{p}}\Delta t$$

### 2.7. Reverse-Osmosis and Pumping Loads

RO operates by forcing brackish water to flow through a membrane under high pressure. Water molecules pass through the membrane to form a flow of fresh water, while ions and other molecules are rejected with wastewater. Two high-pressure pumps are required to maintain constant pressure on the membrane, forming the RO electrical load $P_{RO}$. For an RO unit operating at a fixed production level, the pressure required at the membrane is dependent on the temperature of the incoming brackish water from the storage tank [26]. A higher temperature serves to reduce the pressure required for a fixed production, thus lowering power needs. Figure 4 plots the RO pumping power $P_{RO}$ versus the temperature of the brackish water $T_{T}$ in the storage tank for a constant production of 360 m^3^/day. The RO modeling Equations described in [26], along with parameters from DOW Chemical technical documentation, are used to develop this relationship.

The other significant load component for the system is the power required for pumping water through the PVT and tank from the reservoir source. There will be $N_{p}/2$ small pumps for moving the water through the PVT array. The electrical power required for pumping the brackish water through the PVT array and tank is given by: (17)$$P_{sys}=\frac{\rho gh{\dot{M}}_{F}}{(3.6\times 10^{6})\eta_{HP}}$$
where ρ, g, and h represent water density, gravitational acceleration, and pipe height, respectively. The total electrical power consumption for the plant is the summation of RO power and electrical power required for pumping the brackish water through the system.
(18)$$P_{Total}=P_{RO}+P_{sys}$$

## 3. Electrical Power Dispatching

Referring to Figure 1, the electrical side of the system requires a dispatching algorithm that allocates PVT electrical power $Q_{e}$ to the load and battery, decides when to discharge the battery, and decides when to use grid power. The plant electrical load and the battery storage charge level determines the dispatching process hourly. If the plant electrical load exceeds the PVT electrical output, then all of the PVT electrical power is dispatched to the plant and, if possible, the battery satisfies the remainder. Utilization of grid power occurs if the PVT and battery cannot meet the load. When PVT electrical output exceeds the load, the excess power is stored in the battery.

The PVT electrical output is split at each hour t into energy $E_{2}\left(t\right)$ sent directly to the load, and energy $E_{1}\left(t\right)$ to charge the battery. Therefore,
(19)$$Q_{e}\left(t\right)=E_{1}\left(t\right)+E_{2}\left(t\right)$$

The total electrical load, $P_{total}\left(t\right),$ for the plant must be satisfied at each hour, such that:(20)$$P_{total}\left(t\right)=E_{3}\left(t\right)+E_{2}\left(t\right)+E_{G}\left(t\right)$$
where $E_{3}\left(t\right)$ is the battery output to the load and $E_{G}\left(t\right)$ is the energy purchased from the grid. The battery charge level is B(t), which is updated each hour according to
(21)$$B\left(t\right)=B\left(t-1\right)+E_{1}\left(t\right)-E_{3}\left(t\right)$$

Characterization of the battery storage includes the following three parameters: a maximum storage capacity $B_{H}$, a minimum storage capacity $B_{L}$, and the number of hours h required to charge or discharge the battery. This means that there is a limit on the hourly energy transfer to and from the battery, accordingly:(22)$$E_{3}\left(t\right)\leq \frac{B_{H}-B_{L}}{h},E_{1}\left(t\right)\leq \frac{B_{H}-B_{L}}{h}$$

Additionally, the battery charging and discharging at the same time is not allowable. At each hour, the dispatching proceeds based on Equations (19)–(22) and a few simple rules. If the electrical load exceeds PVT output, then all of the output is dispatched to the system and the battery satisfies as much of the remainder as possible, with grid power used as a last resort. If PVT output exceeds the load, then as much of the excess PVT power as possible is stored in the battery, and any remaining power is sold back to the grid.

### System Simulation

Using the models for the PVT, thermal storage tank, RO, and electrical dispatching, a single dynamic model simulation for the system is constructed. The system is characterized by a fixed production level of 362 m^3^/day of distilled water, a location for the TMY3 data, and four system parameters: $N_{s}$, $N_{p}$, $M_{T}$, and $B_{cap}$. To illustrate the action of the system, plots are included here for a single location in KSA with the following conditions: $N_{s}=10$, $N_{p}=80$, $M_{T}=250,000\;\mathrm{kg}$, and $B_{cap}=1000\;\mathrm{kWh}$. Figure 5 shows the total system electrical load along with the PVT electrical output, indicating the peaks due to solar activity during each day. Also shown is the tank temperature, and it is clear that the load decrease for higher temperatures. The variations in tank temperature are about 5 degrees, causing a power variation of about 10 kW, which represents about 15% of the peak load.

The PVT electrical output is only large enough to meet the full load during part of the day. At other times, either the battery or the grid power makes up the rest of the load. Figure 6 illustrates the power dispatching from each source to the load. This clearly shows a similar daily cycle, where PVT output can meet the load during the sunny hours, the battery then meets the load in the early evening and night hours, and the grid meets the load in the early morning hours after the battery has drained and before the solar activity begins again. 

The battery and PVT array sizes chosen as such that the load (zero curtailed PVT power) can utilize all of the PVT electrical power output. Therefore, it is notable that expanding the size of the PVT system to function completely on renewable energy significantly adds to the system cost.

To get a sense of average system performance, Figure 7 illustrates monthly totals for the year in terms of three categories of power used to meet the load: power directly from the PVT to the load, power indirectly to the load from the PVT via the battery, and grid power. This figure shows that for the system parameters selected, the typical monthly PVT power penetration is about 70%. In other words, about 70% of the load satisfies the renewable power generated by the PVT array. 

The system performance as illustrated by Figure 7 is a function of the PVT array size, the tank volume, and the battery capacity. Average renewable penetration for the year is a single performance measure that can characterize the system as a function of the system parameters. The next section develops an economic model for the system as another measure of performance. This model can be used to cost optimally size the PVT system for a specific application. 

## 4. System Cost Model

This section presents a model for total annual plant cost, represented as the sum of payment towards an amortized loan and annual operating costs (grid-power purchases plus maintenance). Both the loan payment and operating costs are functions of the PVT array size, the battery capacity, and the tank size. The up-front system capital cost functions as an investment repaid via a loan with a fixed interest. Operating costs are determined by simulating for one year. Table 1 lists variables required to determine system cost. It also shows the unit costs (with references) employed for the study [27]. 

The full system capital cost is determined by the following components: PVT array, batteries, RO, storage tank, high-pressure pumps for the RO, and low-pressure pumps for moving water through the PVT. The PVT capital cost computes as the product of the number of panels with a per-panel price. Similarly, the battery capital cost is proportional to its storage capacity $B_{cap}$, and the storage tank capital cost is proportional to its volume. The RO capital cost is proportional to production capacity $RO_{cap}$. All of the individual component costs sum the full capital cost, as follows.
(23)$$\begin{array}{c} CC=CC_{PVT}\cdot N_{s}\cdot N_{p}+CC_{bat}\cdot B_{cap}+CC_{T}\cdot V_{T}+CC_{RO}\cdot RO_{cap}+CC_{NP}\cdot N_{HP}\\ +CC_{LP}\cdot N_{LP} \end{array}$$

KSA encourages and offers incentives to effectively reduce this up-front cost burden, modelled here as a renewable credit RC. Additionally, KSA offers lower interest rates i for financing renewable-energy projects [28]. The entire system is assumed to operate for a lifetime $T_{SYS}$, and the loan is amortized over this period of time, leading to the following loan payment.
(24)$$Annual\;Loan\;Payment=CC\cdot \left(1-RC\right)\frac{i}{1-\frac{1}{{\left(1+i\right)}^{T_{SYS}}}}$$

The annual grid cost is found by running the dispatch model for the full year and summing all of the hourly grid energy purchases $E_{G}\left(t\right)$. Currently, KSA electrical prices are increasing, which serves to further motivate the renewable approach [29]. The grid electrical purchases are computed on a flat price per-kWh basis $P_{gen}$. The total annual cost for the plant is the sum of the annual loan payment, and the grid energy purchases.
(25)Total Cost=Annual Loan Payment+Annual Grid Cost

## 5. Optimization Process

The total cost viewable as a nonlinear objective function, which depends on the PVT array size, the tank volume, and the battery storage capacity. The total cost optimization over these variables, while keeping all other parameters (interest rate, prices, plant location, production level, etc.) constant is a necessity. Furthermore, it is possible to perform cost optimization while constraining the average renewable penetration level. Evaluating the cost objective function requires the following steps.
Input the PVT array size, tank volume, and battery capacity.Compute the hourly PVT useful heat and electrical energy output, tank temperatures, and RO electrical demandOperate the hourly electrical dispatching simulation to determine system energy flows, grid purchases, etc.Determine the average renewable penetrationEvaluate the total system cost using Equations (21)–(25).

The cost objective function is optimized using Matlab’s “*fmincon*” package, which implements sequential quadratic programming to find the two PVT array dimensions, tank volume, and battery capacity to minimize cost with constraints on the renewable penetration. 

## 6. Results

The cost function is minimized when the PVT supplies 70% of the total system electrical energy needs (70% penetration), with conventional grid power supplying the remaining 30%. In this case, the PVT array size is 10 panels in series ($N_{s}=10)$ by 80 panels in parallel ($N_{p}=80)$. The tank volume required is 250 $m^{3}$, and the battery capacity is 1000 kWh. The annual cost for the PVT system is 18% less than the annual cost for driving the plant with 100% conventional electricity and no pre-heating of the feedwater. The production level is fixed at 362 m^3^/day of distilled water. The following sub-sections describe the performance of the individual system components in detail, along with the components of optimized cost.

### 6.1. PVT Performance

The fluid circulation increases the pumped water temperature to improve the system water production. Compared to a PV panel in the same climatic conditions, a PVT panel can produce more electrical energy due to the decrease in cell temperature caused by the coolant circulation. Figure 8 shows plots of hourly temperature for the PVT cells, the output water, and the tank temperature. The tank serves to buffer the output water temperature, to avoid sending overheated water to the RO, which could damage it. The PV cell temperatures reach as high as 80 °C at peak times during the day, which reduces efficiency, but this is within the safe temperature range for their operation. 

The hourly thermal and electric powers from the PVT shown in Figure 9 are on a per-unit-area basis. The PVT generates 3–4 times as much thermal as electric power at peak times. At night, there is no electric power produced, but it is possible to collect thermal energy if the ambient air temperature is higher than the incoming feed-water temperature. This occurs at night during the summer months. 

The thermal efficiency as defined by Equation (8) plotted in Figure 10, along with electrical efficiency as determined by Equation (11). The definition for thermal efficiency only applies during hours when sunlight is incident on the collector. The results show that at peak times of the incoming solar energy converted into useful heat from 60–70%, and that is about 12–16% of the incoming energy converted into electrical energy. The electrical efficiency does drop when the PV temperature is high. Combined peak efficiency for the PVT array is therefore about 80%, which is close to reported values for this type of system [24,30,31]. 

### 6.2. Cost Optimization with Penetration Constraints

The non-linear cost objective function was minimized with constraints on the renewable energy penetration, which allows for identifying the level of penetration that produces the lowest cost, and it allows for a comparison of extreme cases (very low and very high penetration). Figure 11 shows this result and the cost minimization when the PVT supplies 70% of the total system energy needs (70% penetration). 

### 6.3. PVT Load and Cost Reduction

Each cubic meter of purified water produced by the RO system requires a certain amount of electrical energy, depending on the feed-water temperature. Figure 12 compares the monthly energy required per cubic meter for two cases: when the feedwater enters the RO directly with no PVT heating, and for the cost-optimized system at 70% penetration that uses PVT feed-water heating and thermal storage. The average monthly percent reduction in energy needs due to the PVT heating is 16%.

In terms of economic cost, the PVT and additional hardware increase the portion of the annual cost due to payment towards the capital. The benefit for PVT, however, is the lower operating costs, as the PVT reduces the RO electrical energy needs and supplies 70% of that electrical energy. Figure 13 illustrates the monthly average cost per cubic meter of production, for a plant that has no PVT (100% grid power), and a plant that uses the optimized PVT system. The total annual costs for each case divided are between the portion allocated to capital and the operating costs. On average, the PVT system serves to reduce costs by 30% relative to a conventional fossil fuel-powered system with no PVT. The PVT augmented RO system is the cost-optimal option. 

## 7. Conclusions 

In this research, a theoretical approach to building a low energy cost desalination system powered by renewable energy is an optimistic proposal. The research contributes to the knowledge in the design of a low-cost desalination system and that building this in remote areas in the Mediterranean region is beneficial. In particular, this work presents a design for optimizing the integration of PVT into small-scale desalination. This is appropriate for a remote community in KSA, such that the PVT electrical output and conventional grid power will satisfy the electrical load for the plant and the excess PVT electrical power may be stored in a battery for nighttime desalination, along with grid power as needed to maintain continuous RO production. The PVT thermal output can raise the feedwater temperature, improving RO performance. The energy demands for RO reduces by preheating the feed brine. The non-linear cost objective function minimizes with constraints on renewable energy penetration. When the PVT supplies 70% of the total system energy needs (70% penetration), the cost decreases. The dynamic modeling for this study indicates that the optimal PVT, hot water tank, and battery capacities are $N_{s}=10$, $N_{p}=80$, $M_{T}=250\;m^{3}$ and $B_{cap}=1000\;\mathrm{kWh}$, respectively. In this case, on average, the PVT system serves to reduce costs by 30% relative to a conventional fossil fuel-powered system with no PVT. Future studies to validate the potential of the proposed system experimentally is a necessity. Furthermore, other designs could include coupling PVT with CHP, since the CHP can be ramped up or down to complement the PVT output. The design has the potential to optimize the size of each component as a function of climate and location. Additional designs could include coupling PVT with combined MSF and RO, the thermal output from PVT entered into the MSF unit, which will increase the production of MSF. Then, the rejected brine of the MSF unit is hot and split into two streams. One stream is fed into the RO unit and the other stream goes out to the sea. The electricity output from PVT will drive the power consumption for MSF and RO.

## Figures and Tables

**Figure 1 ijerph-18-01001-f001:**
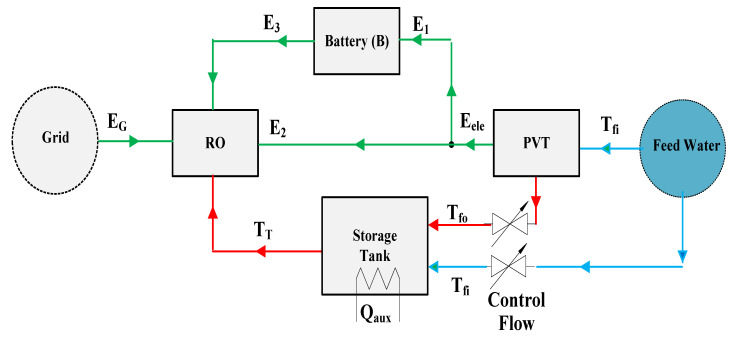
Diagram for PVT- and RO-based technology system for water desalination plant (T refer for thermal and E refer for electrical energy).

**Figure 2 ijerph-18-01001-f002:**
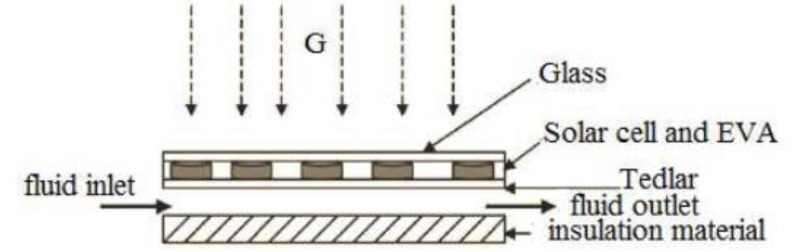
Cross-section of a PVT flat-plate collector.

**Figure 3 ijerph-18-01001-f003:**
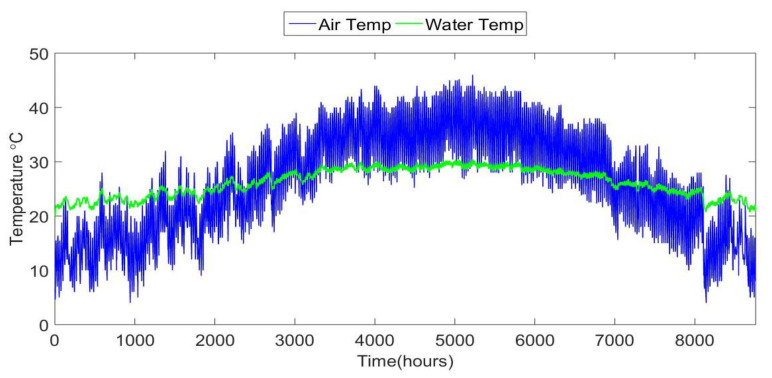
Ambient air temperature from TMY3 data, and water temperature model.

**Figure 4 ijerph-18-01001-f004:**
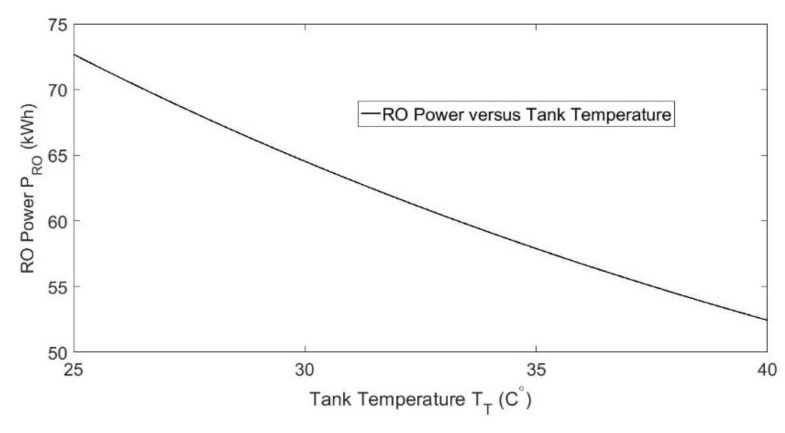
RO pumping power versus brackish water temperature in the storage tank.

**Figure 5 ijerph-18-01001-f005:**
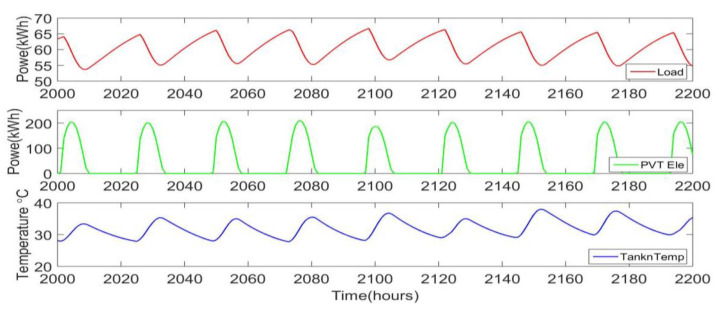
Total system electrical load, PVT electrical output, and tank temperature (bottom).

**Figure 6 ijerph-18-01001-f006:**
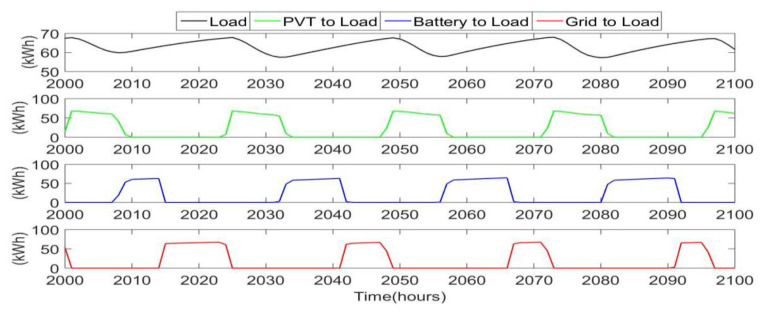
Electrical dispatching to meet the load, including direct power from PVT, stored battery power, and grid power.

**Figure 7 ijerph-18-01001-f007:**
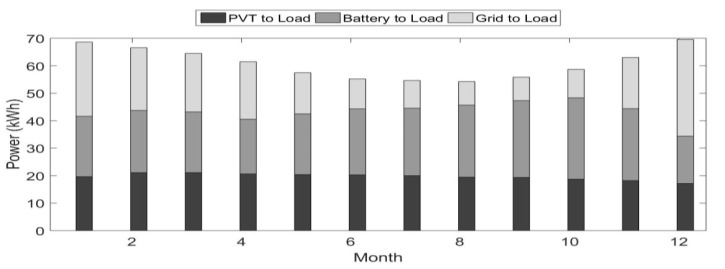
Monthly totals for the year in terms of three categories of power used to meet the load.

**Figure 8 ijerph-18-01001-f008:**
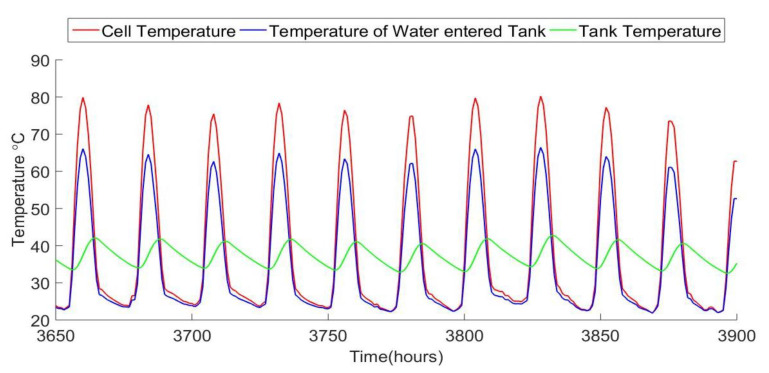
PVT cell temperature and fluid output temperature, along with tank temperature, for several summer days of operation.

**Figure 9 ijerph-18-01001-f009:**
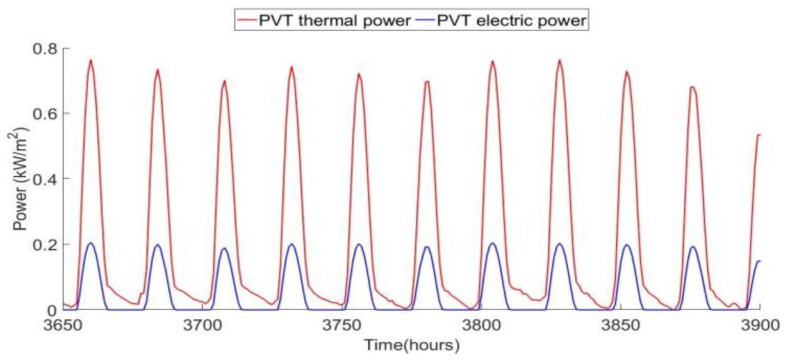
Hourly PVT thermal and electric power production for summer days.

**Figure 10 ijerph-18-01001-f010:**
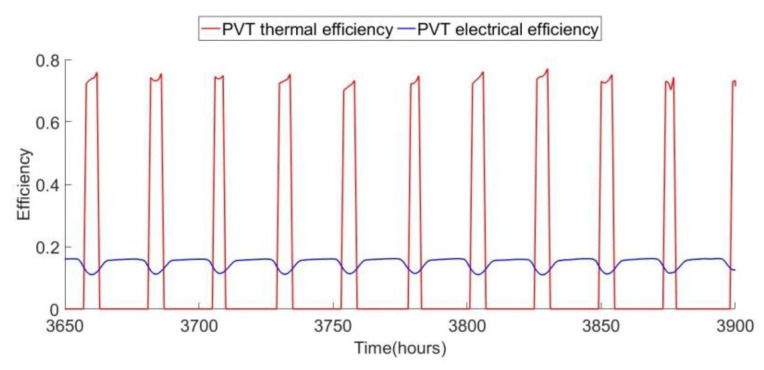
Hourly variations in PVT thermal and electric efficiency.

**Figure 11 ijerph-18-01001-f011:**
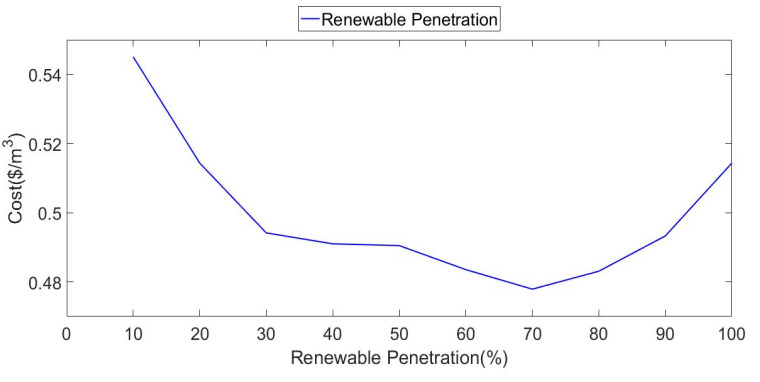
Minimized total annual cost as a function of the penetration constraint.

**Figure 12 ijerph-18-01001-f012:**
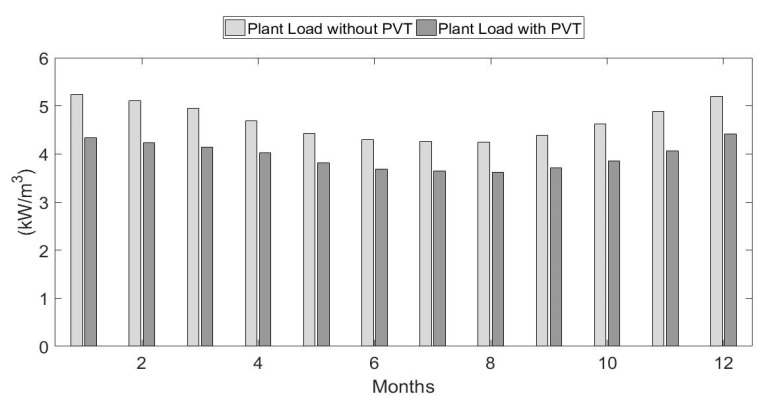
Monthly plant energy loads with and without PVT.

**Figure 13 ijerph-18-01001-f013:**
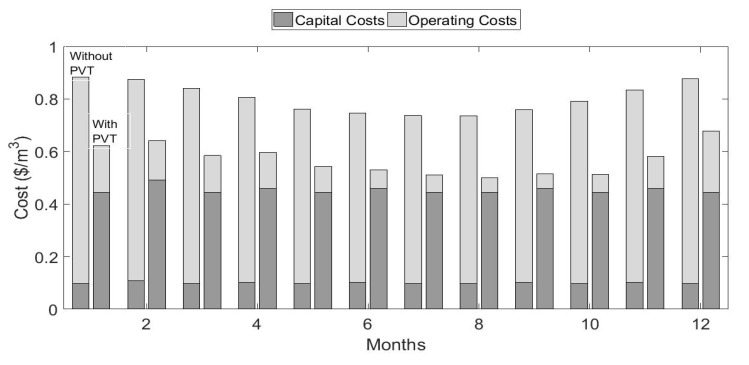
Monthly average cost/cubic meter of production, for a plant that has no PVT versus a plant with optimized PVT.

**Table 1 ijerph-18-01001-t001:** Variables used for calculating annual system cost.

Variable	Description	Variable	Description
CC	Total system capital cost ($)	$CC_{RO}$	RO capital cost ($/m^3^/day)
RC	KSA renewable-energy credit (%)	$RO_{cap}$	RO production capacity (m^3^/day)
$CC_{PVT}$	PVT capital cost ($/panel)	$CC_{HP}$	High-pressure pump capital cost ($/pump)
$N_{C}$	The number of PVT panels in series.	$N_{HP}$	The number of high-pressure pumps.
$N_{P}$	The number of PVT panels in parallel.	$CC_{LP}$	Low-pressure pump capital cost ($/pump)
$CC_{bat}$	Battery capital cost ($/kWh)	$N_{LP}$	The number of low-pressure pumps.
$B_{cap}$	Battery capacity (kWh)	$T_{sys}$	System lifetime (years)
i	Loan interest rate	$CC_{T}$	Storage tank capital cost ($/m^3^)
$P_{gen}$	Grid generation price ($/kWh)	$V_{T}$	Storage tank volume (m^3^)

## Data Availability

Not applicable.

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
