# Peer review of "Design and Energy Requirements of a Photovoltaic-Thermal Powered Water Desalination Plant for the Middle East"

_ijerph, 2021, doi:10.3390/ijerph18031001_

Round 1
Reviewer 1 Report
Review of Alqaed et al. ERPH
This MS is a design of a water desalination plant in Saudi Arabia powered primarily by solar energy. The design using PVT technology appears to be novel. I am not an electrical engineer, so I cannot evaluate the variables and equations used to cost out the electricity produced by the plant. My comments about this MS are more general in nature. The paper is well written and easy to read. I recommend acceptance after dealing with the minor points below. For clarity, I suggest changing the title to Design and energy requirements of a photovoltaic-thermal powered water desalination plant for the Middle East. My overall comment is that there should be a better link between the proposed plant and an existing desalination plant using similar technology. Is the proposed plant based on an existing real plant design but modified according to your assumptions and statements in the introduction? Need a link back to real world desalination plant technology and clarify how much of an advance in the state of the art the new plant will be. Comparisons with these existing plants will allow a true consideration of the proposed technology's feasibility. In the results, need to know value of each variable used to calculate the 70% PVT supply value (I think this is “CC” in equation 23). Minor points: Correct “error reference source not found” throughout by including figure numbers. Line 13: ...conversion. To address these challenges, a small... L15: ...(KSA) was designed and it’s power requirements calculated. This... L30: The current global... L35: ...20 years it is predicted that ...6 million m3 per day [14]. L47, 52: Persian Gulf L99: high ambient and operating temperatures... L110: ...model. An optimization...system is in section 5. L115: ...diagram of the proposed water desalination plant that combines PVT and RO technology. Figure legends need to be more descriptive. Fig 1, spell out or define all abbreviations. Fig 2: what is G? Scale? L134: ...Cross section in figure 2. Table 1: how did you choose these variables? Based on a previous reference? Need to provide this ref. L362: ...close to reported values... : need references to support this statement. m3 to m^3 throughout.Author Response
Please see the attachment

Reviewer 2 Report
The current paper reports on a techno-economic evaluation of a solar-powered small scale desalination unit. The paper is written and structured well. A good overview of the current desalination options is included in the introduction. The study has been carried out well, and all relevant factors have been taken into account. I only have few questions that could be addressed in a minor revision of the manuscript:
- Throughout the text, the references to figures are replaced by error messages. Please correct.
- Only the influence of the temperature on the viscosity and hence the pumping power required is discussed, but what is the effect on the selectivity of the membranes used?
- The option of non-continuously operating the installation was not investigated. Why not only operate the system in the daytime, or as long as the stored solar energy is available via the battery or thermal storage?
- I am missing a sensitivity analysis of the interest rate. A reduced interest rate in KSA is mentioned, but it would be interesting to see what the possibilities are for other countries as well.
I recommend a minor revision of the manuscript.
